# Hypomethylation of ATP1A1 Is Associated with Poor Prognosis and Cancer Progression in Triple-Negative Breast Cancer

**DOI:** 10.3390/cancers16091666

**Published:** 2024-04-25

**Authors:** Yesol Kim, Je Yeong Ko, Hyun Kyung Kong, Minyoung Lee, Woosung Chung, Sera Lim, Dasom Son, Sumin Oh, Jee Won Park, Do Yeon Kim, Minju Lee, Wonshik Han, Woong-Yang Park, Kyung Hyun Yoo, Jong Hoon Park

**Affiliations:** 1Department of Biological Science, Research Institute of Women’s Health, Sookmyung Women’s University, Seoul 04310, Republic of Korea; yesol@wustl.edu (Y.K.); jeyeong@sookmyung.ac.kr (J.Y.K.);; 2Samsung Genome Institute, Samsung Medical Center, Seoul 06351, Republic of Korea; 3Cancer Research Institute, Seoul National University College of Medicine, Seoul 03080, Republic of Korea; 4Department of Surgery, Seoul National University College of Medicine, Seoul 03080, Republic of Korea; 5Department of Health Sciences and Technology, Samsung Advanced Institute for Health Sciences & Technology, Sungkyunkwan University, Seoul 06355, Republic of Korea; 6Department of Molecular Cell Biology, Sungkyunkwan University School of Medicine, Suwon 16419, Republic of Korea

**Keywords:** triple-negative breast cancer, DNA methylation, ATP1A1, prognostic marker

## Abstract

**Simple Summary:**

Triple-negative breast cancer (TNBC) is the most aggressive subtype of breast cancer, but effective biomarkers has yet been developed. Here, we identified ATP1A1 as prognostic marker and therapeutic target via integrated analysis of DNA methylome and transcriptome of TNBC. These data will provide new insight into not only developing specific biomarker for TNBC but also understanding TNBC pathogenesis.

**Abstract:**

Dysregulated DNA methylation in cancer is critical in the transcription machinery associated with cancer progression. Triple-negative breast cancer (TNBC) is the most aggressive breast cancer subtype, but no treatment targeting TNBC biomarkers has yet been developed. To identify specific DNA methylation patterns in TNBC, methyl-binding domain protein 2 (MBD) sequencing data were compared in TNBC and the three other major breast cancer subtypes. Integrated analysis of DNA methylation and gene expression identified a gene set showing a correlation between DNA methylation and gene expression. ATPase Na+/K+-transporting subunit alpha 1 (ATP1A1) was found to be specifically hypomethylated in the coding sequence (CDS) region and to show increased expression in TNBC. The Cancer Genome Atlas (TCGA) database also showed that hypomethylation and high expression of ATP1A1 were strongly associated with poor survival in patients with TNBC. Furthermore, ATP1A1 knockdown significantly reduced the viability and tumor-sphere formation of TNBC cells. These results suggest that the hypomethylation and overexpression of ATP1A1 could be a prognostic marker in TNBC and that the manipulation of ATP1A1 expression could be a therapeutic target in this disease.

## 1. Introduction

Breast cancer is a heterogeneous group of cancers divided into four main categories according to their intrinsic gene signatures: Luminal A (LumA), defined as estrogen receptor (ER) and/or progesterone receptor (PR)-positive and human epidermal growth factor receptor 2 (HER2)-negative; Luminal B (LumB), defined as ER- and/or PR-positive and HER2-positive or -negative; HER2-overexpressing; and triple-negative breast cancer (TNBC), lacking ER, PR, and HER2 expression [1]. TNBC accounts for approximately 15–20% of all newly diagnosed breast cancers, and often shows recurrence and aggressive growth due to acquisition of chemo-resistance, despite chemotherapy being initially effective [2,3]. These features result in an overall poor prognosis, characterized by higher rates of disease recurrence and distant metastasis than in patients with other breast cancer subtypes [4]. Although TNBC is a major cause of mortality in patients with breast cancer, therapeutic options for advanced TNBC remain limited. Therefore, the identification of novel therapeutic strategies is necessary.

Epigenetic mechanisms of gene regulation are heritable reversible modifications that mediate tissue-specific gene expression [5,6,7]. DNA methylation is a common epigenetic marker involving the modification of cytosine residues in CpG dinucleotides by methyl groups [8,9,10]. DNA methylation is crucial for proper development and differentiation in humans [11] and has long been associated with the establishment and maintenance of imprinted gene expression and X-chromosome inactivation [8]. Although DNA methylation is critical for cancer development, its role in breast cancer subtypes, particularly TNBC, is not yet well understood. DNA methylation has been profiled in TNBC, but the genome-wide localization of methylated DNA in this and other breast cancer subtypes remains to be determined. Additionally, integrative studies of DNA methylation and mRNA expression are required to provide insights into the potential biological activities of epigenetic alternations.

ATP1A1, encoding the Na^+^/K^+^-ATPase α1 subunit, is a member of the P-type ATPase family. This Na^+^/K^+^-ATPase is an ion pump that uses energy from ATP hydrolysis and is located ubiquitously in membranes, transmitting signals to intracellular compartments [12,13]. ATP1A1 has been reported to be overexpressed in breast cancers, but the methylation status and biological function of ATP1A1 in TNBC have not yet been elucidated [14]. Cardiac glycosides (CGs), such as digoxin, which are used to treat heart failure, have been found to have anti-cancer activity [15]. Indeed, the inhibition of ATP1A1 using CGs was reported to be an option for cancer therapy [16,17]. The present study was designed to identify DNA methylation patterns specific to TNBCs and to identify specific methylation markers in these tumors, allowing the design of targeted treatments for TNBC.

## 2. Materials and Methods

### 2.1. Breast Cancer Tissue

Fresh-frozen human tissue samples, consisting of breast tumors and adjacent non-tumorous regions of each breast cancer subtype (Luminal A, Luminal B, HER2+, and TNBC), were obtained from patients undergoing surgical resection at Seoul National University Hospital. Informed consent was obtained from all patients before collection of the specimens. This study was approved by the Institutional Review Board of Seoul National University Hospital (IRB number: 1704-019-843).

### 2.2. MBD-Seq Library Preparation and Next-Generation Sequencing

Genomic DNA was isolated from normal breast and breast cancer samples with Qiagen Puregene kits (Qiagen, Valencia, CA, USA) following the manufacturer’s protocol. Approximately 1 µg genomic DNA was sheared to 200 to 400 bp fragments using a Covaris LE220 ultrasonicator (Covaris, Inc., Woburn, MA, USA). The sonicated genomic DNA was immunoprecipitated with human MBD2 protein, coupled to paramagnetic Dynabeads M-280 Streptavidin via a biotin linker using a MethylMiner Methylated DNA Enrichment Kit (Invitrogen, Carlsbad, CA, USA) according to the manufacturer’s protocol. The methylated fractions were purified as a single enriched population with an elution buffer containing 2M NaCl. The methylated double-stranded (ds) DNA was end-repaired, and ‘A’ was ligated to the 3′ end. Illumina adapters were ligated to the fragments, and DNA libraries were constructed from 500 ng of MBD2-enriched DNA for each breast cancer subtype, according to the Illumina protocol.

### 2.3. Data Analysis of MBD-Seq

Sequenced reads were aligned to the human genome reference (hg19) using bowtie [18,19], allowing three mismatches; read maps to multiple genomic positions were excluded. Using the MEDIPS R package, the raw counts from MBD-seq for classified regions of interest (ROIs), especially CpG- and non-CpG-coding DNA sequences (CDSs) and promoter regions, were normalized to yield absolute methylation scores (ams) [20]. For each ROI matrix, the ams values in each sample were normalized to a range of 0–1. Hypermethylated and hypomethylated genes were defined by differences among means of tumor and normal samples, and the genes with fold changes > 0.2 were classified as hypermethylated, and genes with fold changes < −0.2 were classified as hypomethylated. Features of CpG and non-CpG islands were compared by extracting the 1000 most variable genes estimated by SD/SD_max_ across all samples from each ROI matrix. Genomic DNA data can be accessed at GSE116877.

### 2.4. RNA Preparation and mRNA Microarray

Total RNA was isolated using TRIzol reagent (Invitrogen, Austin, TX, USA), followed by extraction with chloroform and precipitation with isopropanol. For quality control, RNA purity and integrity were evaluated by denaturing gel electrophoresis and measuring their OD_260/280_ ratios. Total RNA was amplified and purified using an Illumina TotalPrep RNA amplification kit (Ambion) to yield biotinylated cRNA according to the manufacturer’s instructions. A 750 ng aliquot of labeled cRNA was hybridized to each HumanHT-12 v.2 expression bead array for 16–18 h at 58 °C (Illumina, Inc., San Diego, CA, USA), with array signals detected using Amersham fluorolink streptavidin-Cy3 (GE Healthcare Bio-Sciences, Little Chalfont, UK) and scanned with an Illumina BeadArray Reader confocal scanner. Raw data were extracted using Illumina GenomeStudio v2011.1 software (Gene Expression Module v1.9.0). Probe signal values were log-transformed and normalized using the quantile method. mRNA microarray data can be accessed at GSE113865.

### 2.5. Gene Ontology (GO) Analysis

Gene Ontology (GO) was assessed by uploading the gene list onto DAVID 6.8 (https://david.ncifcrf.gov/, accessed on 5 April 2018), with annotation and visualization according to standard protocols [21]. GO was analyzed statistically using the overrepresentation test with Bonferroni correction for multiple testing.

### 2.6. Acquisition of TCGA, GEO Data, and Analysis of Survival and Its Association with Molecular Subtype

DNA methylation and RNA expression were analyzed in combination using clinically available datasets from the TCGA breast cancer (BRCA) cohort. Raw HM450K methylation data (Level 1) and processed RNA-seq expression data (level 3) were downloaded from the TCGA data portal. Only overlapping samples between the two platforms were obtained. The 764 samples included 108 Luminal A, 46 Luminal B, 14 HER2-enriched, 41 basal-like, and 5 normal-like samples, as well as 549 classified as unknown because of missing clinical information. Survival relative to three genes was analyzed using the R Survival package [22]. In assessing DNA methylation, the 764 samples were partitioned into two groups relative to the median gene methylation of each gene. This criterion was applied to the RNA expression dataset. In the combination dataset, the criteria defining high risk differed for the three genes: hypomethylation and high expression for ATP1A1, hypermethylation and low expression for ADH1C, and hypomethylation and low expression for CFH. Other combinations were classified as low risk. Kaplan–Meier survival curves of these two groups were compared by log-rank tests.

For analysis of gene expression across the breast cancer subtypes, METABRIC and GSE38959 were analyzed. Overall survival (OS) relative to the HM450K probe was calculated using survminer R packages. The correlations of ATP1A1 expression with relapse-free survival (RFS) and distant metastasis-free survival (DMFS) in patients with basal-like breast cancer who did and did not receive chemotherapy were assessed by the Kaplan–Meier method and compared by the log-rank test [23]. Subtype-specific ATP1A1 mRNA expression analysis in the TCGA breast cancer dataset was performed using cBioportal [24,25,26], with relative expression defined as the standard deviation from the mean of expression in the reference population (Z-score).

### 2.7. Bisulfite Treatment and MS-HRM Analysis

The EZ DNA Methylation-Gold^TM^ Kit (Zymo Research, Orange, CA, USA) was used for bisulfite conversion of genomic DNA, according to the manufacturer’s instructions. MS-HRM was performed and quantified as described previously using primer sets designed by Sequenom (http://www.epidesigner.com, accessed on 5 April 2018), as listed in Appendix A. Bisulfite-converted genomic DNA was amplified with each MS-HRM primer set using EpiTect HRM™ PCR kits (Qiagen, Valencia, CA, USA). High-resolution melting curve peaks were identified and analyzed by LightCycler 96 Instrument (Roche). Melting peaks were quantified by determining the area under the curve (AUC) using Image J software (version 1.53g), and the percent of DNA methylation was calculated by linear regression between AUC values and methylation standards of 0%, 50%, and 100%.

### 2.8. Cell Culture and Transfection

SK-BR-3, T47D, MDA-MB-468, and MDA-MB-157 cell lines were purchased and authenticated from the American Type Culture Collection (ATCC^®^, Manassas, VA, USA). MCF7, ZR-75-1, BT-20, HCC-1937, and MDA-MB-231 were purchased and authenticated from the Korean Cell Line Bank (KCLB^®^, Seoul, Republic of Korea). Human breast cancer cells were grown in DMEM (SK-BR-3, MDA-MB-157, MDA-MB-468, and Hs578T) or RPMI (MCF7, T47D, ZR-75-1, BT-20, HCC-1937, and MDA-MB-231) with 10% fetal bovine serum (FBS) (Gibco, Waltham, MA, USA) and 0.2% MycoZap™ Plus-CL (Lonza, Portsmouth, NH, USA) in a 37 °C humidified incubator with a 5% CO_2_ atmosphere. Cells were seeded on 10 cm dishes for RNAi transfection. Two *ATP1A1* siRNAs and scrambled siRNA were purchased from Ambion and transfected into seeded cells using Lipofectamine RNAiMAX (Invitrogen, Waltham, MA, USA). For plasmid transfection, the ATP1A1 CDS was cloned into the pCMV-Tag2B vector and transfected into the cells using FuGene (Promega, Madison, WI, USA). Cells were harvested for further analysis and other experiments.

### 2.9. Western Blotting Analysis

For immunoblot analysis, cells were harvested and lysed in RIPA buffer (Thermo Fisher Scientific, Waltham, MA, USA) containing protease inhibitors (Roche, Basel, Switzerland), phosphatase inhibitors (Sigma, Burlington, MA, USA) and PMSF (Sigma). Proteins were isolated and their concentrations measured using bicinchoninic acid (Sigma) and copper (II) sulfate (Sigma) solutions. Primary antibodies against ATP1A1 were purchased from Abcam (Cambridge, MA, USA) and primary antibodies against β-actin (loading control) were from Bethyl Laboratories (Montgomery, TX, USA). Immunoreactive proteins were detected using horseradish peroxidase-conjugated secondary antibodies and the enhanced chemiluminescence reagent, EzWestLumi plus (ATTO, Tokyo, Japan), and subsequently analyzed with an LAS-3000 image analyzer (Fujifilm Life Science, Cambridge, MA, USA). The uncropped blots and molecular weight markers are shown in Appendix A.

### 2.10. Immunohistochemical Analysis of Tissue Microarrays (TMAs)

Paraffin-embedded tissue microarray (TMA) slides, BR1201, were purchased from US Biomax (Rockville, MD, USA). The slides were de-paraffinized, followed by antigen retrieval in sodium citrate buffer, pH 6.8 (ScyTek Laboratories, Logan, UT, USA), for 20 min in a microwave oven. TMA specimens were probed with an anti-ATP1A1 antibody (Abcam, USA) using the VECTASTAIN ABC kit (Vector Labs, Newmark, CA, USA), with staining visualized with a NovaRED substrate (Vector Labs), followed by counterstaining with hematoxylin. The stained slides were subsequently scanned using a ZEISS Axio Scan Z1 microscope (Carl Zeiss Microscopy, Oberkochen, Germany), with staining intensity calculated by the IHC profiler [27] plug-in in Image J software.

### 2.11. Cell Viability Assay

Cells seeded in 10 cm dishes were transfected with control or ATP1A1 siRNAs. After 24 h, the cells were re-seeded in 24-well plates. Cell viability was measured every 24 h for 4 days, and the growth medium of the remaining cells was replaced with fresh culture medium each day. Alternatively, cells seeded in 24-well plates for 24 h were treated with digoxin or DMSO. At the designated time points, the relative numbers of viable and proliferating cells were determined by WST-8 assays (Enzo Life Sciences, Farmingdale, NY, USA). Absorbance at 450 nm was measured using a multi-well spectrophotometer (Synergy HTX, BioTek Instruments, Winooski, VT, USA).

### 2.12. Mammosphere Formation

After siRNA transfection for 24 h, cells were re-seeded at approximately 1 × 10^4^ cells/mL in ultra-low-attachment 6-well plates (Corning Costar, Corning, NY, USA), and grown in suspension with serum-free DMEM/F12 medium (WelGENE) containing B27 (Gibco), 20 ng/mL epidermal growth factor (ProSpec, Rehovot, Israel), 20 ng/mL basic fibroblast growth factor (Gibco), 5 mg/mL insulin, 0.4% bovine serum albumin, and heparin (Sigma-Aldrich). Where indicated, digoxin was added to the medium after 9 days. After 10 days, mammospheres were visualized and counted using a microscope (Olympus IX71). 5-aza-2′-deoxycytidine (Sigma-Aldrich) was used at a concentration of 5 uM in a mammosphere formation assay.

### 2.13. Colony Formation Assay

Cells were seeded into 6-well plates (1000 cells/well) and maintained for 10-14 days. After colony formation, cells were fixed with fixation solution (acetic acid/methanol 1:7) and stained with 0.5% crystal violet.

### 2.14. Apoptosis Analysis

The cells were trypsinized and washed with cold PBS and then stained with FITC and PI by FITC Annexin V apoptosis Detection Kit I (BD Pharmingen, San Diego, CA, USA). Stained cells were analyzed by a flow cytometer (FACS Canto II, BD BioSciences, Franklin Lakes, NJ, USA). The apoptosis rate was calculated by analyzing early- (annexin V-positive) and late-apoptotic (annexin V and PI-positive) cells.

### 2.15. Statistical Analysis

All data were analyzed and differentially expressed genes visualized using R statistical language v.3.3.1 software. Hierarchical clustering was analyzed using the h-clust function in the ‘average’ clustering method and the Euclidean distance function. Differences in gene expression were determined by comparing fold changes using paired t-tests, with the null hypothesis stating that there were no differences between the two groups. The false discovery rate was controlled by adjusting the *p*-value using the Benjamini–Hochberg algorithm. The *p*-values from the melting curve analysis were calculated using paired *t*-tests, and 95% confidence intervals were calculated with GraphPad Prism 5. A *p*-value < 0.05 was considered statistically significant.

## 3. Results

### 3.1. Identification of TNBC-Specific DNA Methylation

To identify differentially methylated regions (DMRs) in TNBCs, the DNA of breast tumors and matched normal samples for each of the four breast cancer subtypes was enriched with methyl-binding domain protein 2 (MBD-2), followed by massive parallel sequencing (MBD-seq). The total number of sequencing reads ranged from 22 to 28 million, as summarized in Appendix A. As a control, the sequencing results for a region of the imprinted *PURG* gene were confirmed, with all samples showing 100% methylation of *PURG* (Appendix A). To identify differentially methylated genes, we performed an exploratory two-dimensional (2D) hierarchical clustering analysis of the methylation peaks across breast cancer subtypes by assessing the associations of methylation peaks with CpG islands (CGIs) in promoters and coding sequences (CDSs) (Figure 1A). Using our criteria of a *p*-value < 0.05 and a delta mean difference of 20%, we determined whether genes in the breast cancer subtypes were hypermethylated or hypomethylated (Figure 1B). We found that more genes in the Luminal A/B and TNBC subtypes were hypermethylated than hypomethylated, whereas more genes in the HER2-positive subtype were hypomethylated. In TNBC samples, 682 genes were hypomethylated and 840 were hypermethylated. In addition, approximately 90% of the hypomethylated peaks in TNBCs were located in non-CGIs, with 62% present in CDS regions (Figure 1C). DNA methylation in TNBC and other subtypes showed a greater degree of alteration in the highest portion of non-CGI and CDS regions than in CGIs and promoters (Appendix A). Because many changes in DNA methylation were distributed in CGI/non-CGI and promoter/CDS regions, further analysis was performed to assess the relationship between DNA methylation and gene expression in all regions including non-CGIs and CDSs. To identify TNBC-specific methylated genes, we compared methylated genes among breast cancer subtypes. Although only 5 genes were shared among subtypes, 468 genes were specifically hypomethylated in TNBC. Of hypermethylated genes, 10 overlapped and 353 were significantly hypermethylated only in TNBC samples (Figure 1D), suggesting that TNBC-specific alterations in DNA methylation may play a critical role in the aggressiveness of these tumors.

### 3.2. Integrative Analysis of DNA Methylation and Gene Expression in TNBC

To better understand the relationships between TNBC-specific changes in methylation and concomitant changes in gene expression, we performed an integrated analysis of gene expression and genome-wide DNA methylation profiles in TNBC samples. We identified 50 genes that were significantly hypomethylated and differentially expressed (|fold changes| ≤ 1.2) and 78 genes that were coordinately hypermethylated and differentially expressed in TNBC (Figure 2A,B). We subdivided differentially methylated and expressed genes into four groups: hypomethylated and upregulated, hypomethylated and downregulated, hypermethylated and upregulated, and hypermethylated and downregulated (Figure 2C). Specifically, we observed 31 genes that were highly expressed and hypomethylated, including genes associated with tumor cell proliferation and migration, such as *ATP1A1*, *EPCAM*, *TOP2A*, and *KIF2C*. Gene Ontology analysis, using DAVID functional annotation tools, showed that genes with deregulated methylation and expression encoded proteins in ATP binding, protein binding, ATPase activity, and actin binding (Figure 2D). Specifically, these genes included those involved in ATP and protein binding, such as *NEK2*, *SOX2*, *CDC7*, and *ATP1A1*.

To determine whether the genes belonging to each of the four groups showed the same TNBC-specific methylation and mRNA expression patterns in large groups of patients, we screened the methylation and RNA expression patterns of selected candidate genes using raw HM450 methylation array data and processed RNA-seq expression data from the Cancer Genome Atlas Network (TCGA) breast cancer (BRCA) cohort. The TNBC-specific methylation and mRNA expression patterns of the *ATP1A1*, *ADH1C*, and *CFH* genes were also significantly altered in the TCGA cohort, depending on the breast cancer subtype (Figure 2E). These findings confirmed that these three genes have the most TNBC-specific expression patterns of all candidate genes.

### 3.3. Strong Association between Hypomethylation/High Expression of ATP1A1 and Poor Survival

To determine whether the *ATP1A1*, *ADH1C,* and *CFH* genes have prognostic value in TNBC, we assessed the relationship between survival and methylation status in the 764 TCGA samples. We found that overall survival (OS) was significantly shorter in patients with hypomethylated than with highly methylated *ATP1A1* (*p* = 0.00834; Figure 3A). In contrast, OS was not significantly associated with the methylation patterns of *ADH1C* and *CFH* in TNBC (Appendix A). Kaplan–Meier survival analysis in patients classified as high risk according to their methylation beta values and gene expression indicated that OS in patients with basal-like breast cancer was significantly poorer when *ATP1A1* was both hypomethylated and upregulated (*p* = 0.00159; Figure 3B).

As the CDS region of the *ATP1A1* gene was specifically hypomethylated in the MBD-seq data, we assessed samples using the HM450K probe, which is specific to the *ATP1A1* CDS region. Strikingly, *ATP1A1* was hypomethylated in two specific regions, cg11756509 and cg10891259, of the basal-like subtype compared with the other subtypes (Luminal A, Luminal B, HER2+, normal-like) (Figure 3C). In addition, we observed that OS was associated with methylation in these two regions. In the basal-like subtype, hypomethylation at these two sites (*p* = 0.025 and *p* = 0.002, respectively) was significantly associated with poor OS, suggesting that this *ATP1A1* hypomethylation was a prognostic biomarker in patients with TNBC (Figure 3D). Moreover, enhanced *ATP1A1* expression was significantly associated with reduced relapse-free survival in patients with the basal-like subtype (*p* < 0.0001; Figure 3E). In addition, high expression of ATP1A1 was associated with poor relapse-free survival (*p* = 0.0027) and poor distant metastasis-free survival (*p* = 0.046) in patients with the basal-like subtype, despite treatment with chemotherapy (Figure 3F). Taken together, these clinical data indicated that integrated methylation and gene expression of *ATP1A1* could be a potential prognostic marker in patients with TNBC.

### 3.4. Validation of Methylation Status and Expression Levels of ATP1A1 in Patients with TNBC

To determine whether the methylation and expression of *ATP1A1* were consistent in databases and primary patient tissues, we compared the extent of *ATP1A1* methylation and expression in TNBC samples and matched normal tissue. Our MBD-seq data showed that exon 19 of *ATP1A1* was specifically hypomethylated. This region was equivalent to the CDS region (Figure 4A) and to the region bound by the TCGA HM450K probe. The extent of methylation in selected regions was analyzed by methyl-specific high-resolution melting (MS-HRM), followed by an estimation of the area under the *ATP1A1* melting curve. The percent of *ATP1A1* methylation in exon 19 was significantly lower in TNBC than in NTL samples (Figure 4B) and was significantly lower in TNBC than in non-TNBC cells (Figure 4C).

We also found that ATP1A1 was highly upregulated in TNBC compared to non-TNBC cell lines (Figure 4D). Expanded analysis of the METABRIC and GEO (GSE38959 [28]) datasets indicated that ATP1A1 is upregulated in basal-like subtype and TNBC tumor samples (Figure 4E). Finally, immunohistochemical analysis of ATP1A1 protein expression in a breast tissue microarray (TMA) that included 120 specimens of different breast cancer subtypes showed that ATP1A1 was more highly overexpressed in TNBC than in other breast cancer subtypes (Luminal A, B, and HER2+) (Figure 4F). Collectively, these findings suggest that ATP1A1 plays a more oncogenic role in TNBC than in other breast cancer subtypes.

### 3.5. Inhibition of Tumor Aggressiveness by Depletion of ATP1A1 in TNBC

ATP1A1 encodes the Na^+^/K^+^-ATPase α1 subunit, a member of the P-type ATPase family, and Na^+^/K^+^-ATPases have been reported to play roles in cell junctions, attachment, motility, and signal transduction [29,30,31,32,33]. To examine the oncogenic role of ATP1A1 in TNBC, we tested the effect of ATP1A1 knockdown on the growth of TNBC cells. The viability of MDA-MB-231 cells transfected with ATP1A1 siRNAs was notably lower than that of cells transfected with non-targeted control siRNA (Figure 5A, upper). Both ATP1A1 siRNAs inhibited the expression of ATP1A1, as shown by Western blotting (Figure 5A, bottom). We also assessed the effect of ATP1A1 knockdown on the self-renewal potential and clonogenic survival of TNBC cells, finding that ATP1A1 siRNAs reduced the number and area of tumor spheres as well as colony formation ability (Figure 5B,C). Furthermore, we confirmed the inhibition of ATP1A1-induced apoptotic cell death of TNBC cells compared to control (Figure 5D). To confirm whether methylation status could affect ATP1A1 expression and self-renewal potential in TNBC cells, we treated 5-aza-2′-deoxycytidine (5-aza-dC), a demethylating agent in MDA-MB-231. 5-aza-dC increased ATP1A1 expression and tumor-sphere area compared to the control (Appendix A), indicating that ATP1A1 expression is regulated by methylation status and that high expression of ATP1A1 increases TNBC tumor spheres.

Cardiac glycosides (CGs), such as digoxin, have been reported to functionally inhibit Na+/K+-ATPases containing ATP1A1 [34]. CGs are clinically approved for the treatment of cardiovascular diseases [15], and it has been reported that digoxin also has anti-cancer effects [16,17]. Therefore, by verifying whether digoxin exhibits antitumor effects on TNBC, we tried to expand the application of digoxin to diseases ranging from cardiac disease to TNBC. We examined the effects of digoxin on MDA-MB-231 cell viability and tumor-sphere formation and found that digoxin reduced MDA-MB-231 cell viability (Figure 6A) and cancer stemness (Figure 6B). We also found that digoxin inhibited the clonogenic survival of TNBC cells (Figure 6C) and increased the proportion of apoptotic cells (Figure 6D). To confirm whether the inhibitory effect of digoxin could be counteracted by ATP1A1 overexpression, we confirmed the changes in digoxin’s effect in ATP1A1-transfected cells compared to control. It was confirmed that ATP1A1 overexpression increased the tumor-sphere area, and the effect of digoxin was reduced in cells in which ATP1A1 was transfected compared to cells in which it was not overexpressed (Appendix A). These data support the anti-cancer effect of digoxin via ATP1A1 inhibition in TNBC cells. Taken together, these data suggest that the inhibition of ATP1A1 expression may impair the progression of TNBC.

## 4. Discussion

Genome-wide profiling studies have investigated alterations in DNA methylation and gene expression in breast cancer [35,36,37,38]. In particular, studies have attempted to identify prognostic markers in TNBCs by identifying significant DMRs in these tumors [39,40]. This study evaluated genome-wide DNA methylation, an epigenetic controller of gene regulation, in TNBCs by using MBD-seq and compared these changes with those observed in other breast cancer subtypes. Additionally, a direct comparison of our MBD-seq and mRNA microarray data with the DNA methylation and expression profiles in TCGA showed that differential methylation of ATP1A1 was critical in TNBC and that the methylation of these genes may be associated with their expression and TNBC progression. Interestingly, we demonstrated that ATP1A1 methylation status affected the outcomes of patients in the TCGA and GEO datasets.

Patients with ER-negative breast cancer, including TNBC, show relatively hypomethylated CpG islands [41]. However, a large-scale analysis of the DNA methylome in cancer reported that gene body methylation has clinical implications [42,43]. To distinguish DNA methylation events of potential functional significance from those that do not biologically contribute to tumorigenesis, we integrated DNA methylation and gene expression profiles of the same tumor and NTL samples. We found that 31 genes were both hypomethylated and upregulated and that alterations in methylation occurred primarily in non-CGI and CDS regions rather than in CGIs and promoters. Notably, we demonstrated that ATP1A1 was hypomethylated in the CDS region, known as the CpG-poor gene body region. Although the precise mechanism by which methylation of the CDS region affects gene expression is unknown, our integrated analyses of methylation and mRNA expression suggest that the hypomethylation of ATP1A1 in the gene body, especially in exon 19, may be a specific biomarker for TNBC.

Studies on the clinical significance of ATP1A1 expression have also been reported in other cancer models. In gastric cancer, ATP1A1 expression shows a correlation with histological type, pathological T stage, and intravascular invasion, and high expression of ATP1A1 significantly reduces the overall survival rate of patients [44]. Moreover, Zhuang et al. found that ATP1A1 was increased in hepatocellular carcinoma (HCC) and its downregulation inhibited the tumorigenesis of HCC cells in vivo [45]. This evidence supports that ATP1A1 has potential as a prognostic marker and a therapeutic target in various types of cancer.

In addition to showing that ATP1A1 was overexpressed in breast cancer, particularly in TNBC, we found that the inhibition of ATP1A1 expression markedly reduced the proliferation and self-renewal activity of human TNBC cells in vitro. Other CGs have been evaluated as anti-cancer drugs for their ability to inhibit tumor growth and enhance the efficacy of radiotherapy and chemotherapy [15,46,47]. For example, a combination of conventional anti-cancer drugs effectively prevented relapses in a TNBC xenograft model [48], and an MAP kinase inhibitor and CGs synergistically induced cell death by targeting ATP1A1 in a mouse melanoma model [49]. These findings indicate that ATP1A1 plays an oncogenic role in TNBC development and is a potential target for TNBC treatment involving CGs such as digoxin.

Taken together, our data indicate that hypomethylation and increased expression of ATP1A1 show poor prognosis in TNBC. Furthermore, the inhibition of ATP1A1 expression decreases TNBC progression in vitro, implying that ATP1A1 is a potential prognostic marker and therapeutic target for TNBC. Further studies are needed for specific targeting of ATP1A1 for the development of therapeutic strategies for TNBC.

## 5. Conclusions

In conclusion, these findings suggest that ATP1A1 methylation and expression may be a useful specific prognostic marker for TNBC and that altered genome-wide methylation in TNBC may be critical for tumor progression. Targeting DMRs in patients with TNBC may therefore be useful in its diagnosis and treatment.

## Figures and Tables

**Figure 1 cancers-16-01666-f001:**
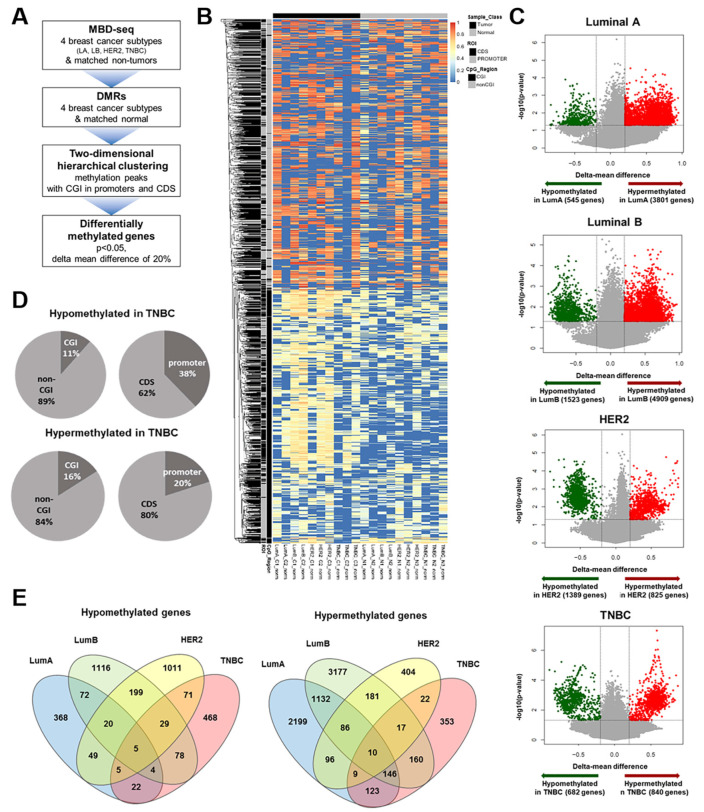
Differences in DNA methylation between breast tumors and NTLs. (**A**) Workflow showing the process of identifying TNBC-specific DNA methylation (**B**) Heat map display of methylation profiles of 10 breast tumor and matched non-tumor samples. Methylation profiles are shown for CpG and non-CpG islands of promoters and CDSs. Columns indicate samples, and rows indicate genes. Red and blue colors represent high and low levels of methylation, respectively. (**C**) Volcano plot of –log10 (*p*-value) versus the delta mean value for differences in methylation between each breast cancer subtype and NTL (non-tumor lesion). Genes that exhibited methylation level differences < 20% are shown in grey. Dashed lines indicate cut-offs for significance. (**D**) Pie charts representing the percentages of hypermethylated and hypomethylated DMRs that overlapped with promoter and CDS or CGI and non-CGI in TNBCs. (**E**) Venn diagram illustrating hypomethylated (**left panel**) and hypermethylated (**right panel**) genes in each breast cancer subtype compared with NTLs ([delta mean] ≥ 2 and *p* < 0.05).

**Figure 2 cancers-16-01666-f002:**
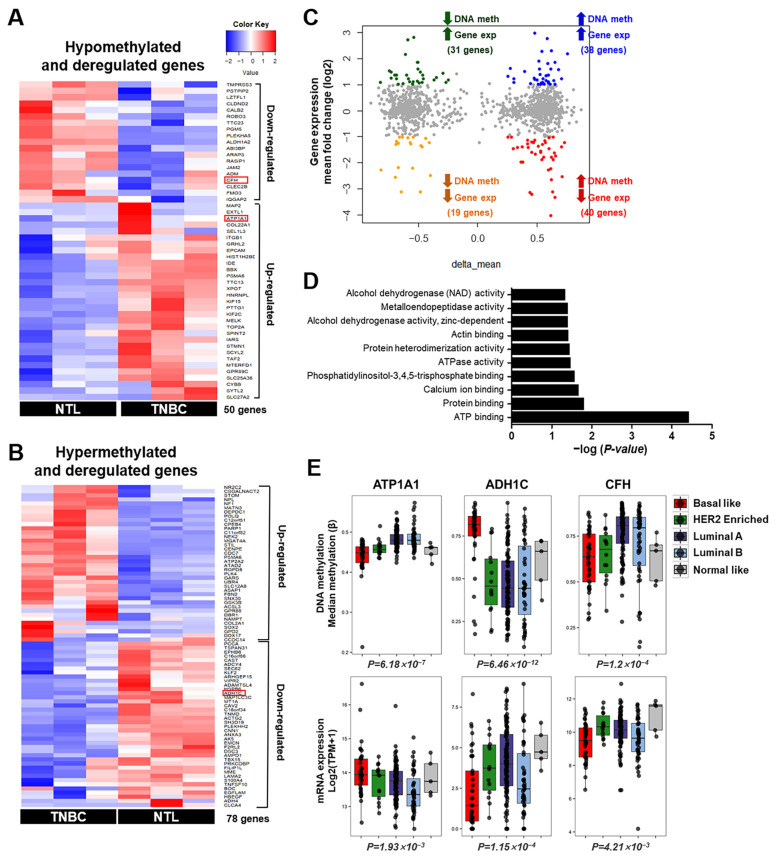
Association between dysregulated genes and methylation changes in triple-negative breast cancer. (**A**) Clustering heat map of the extent of expression of mRNA microarray transcripts for hypomethylated and dysregulated genes in TNBC and matched NTL. (**B**) Hypermethylated and dysregulated genes described in the clustering heat map of expression intensities of mRNA microarray transcripts in TNBC and NTL. (**A**,**B**) Columns indicate samples, and rows indicate genes. Red and blue represent transcripts upregulated and downregulated, respectively, in TNBC relative to NTL. Red boxes indicate genes selected for further analysis. (**C**) Quadrant plot of DMRs and expression of associated genes. The x-axis shows the delta mean values for differentially methylated genes (*p* < 0.05); the y-axis shows the -log2 [fold change] of differential expression for the associated genes. The vertical threshold was [delta mean] ≥ 2, and the horizontal threshold was log2 [fold change] ≥ 1. The four quadrants show genes (i) hypermethylated and upregulated in TNBC (blue circles), (ii) hypermethylated and downregulated in TNBC (red circles), (iii) hypomethylated and upregulated in TNBC (green circles), and (iv) hypomethylated and downregulated in TNBC (orange circles). (**D**) Enriched Gene Ontology analysis of target genes following gene sets. (**E**) Box plots showing the distributions of methylation (**upper panel**) and expression (**lower panel**) levels in each breast cancer subtype. Each box plot shows the median and upper and lower quartiles; whiskers indicate a 1.5× interquartile range; and outliers are marked as dots.

**Figure 3 cancers-16-01666-f003:**
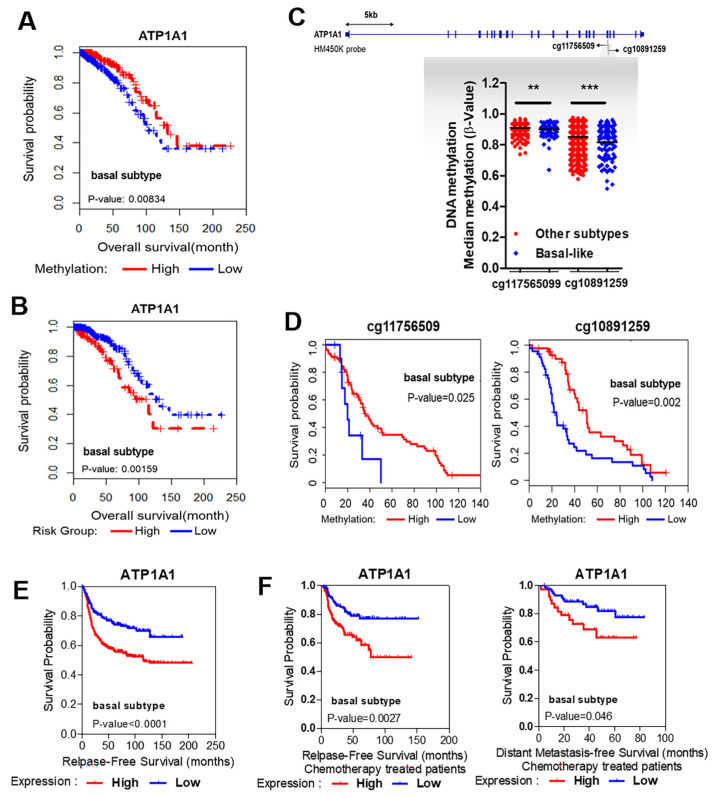
Hypomethylated and upregulated ATP1A1 is associated with poor prognosis in patients with basal-like breast cancer. (**A**) Relationship between *ATP1A1* methylation and overall survival in the basal-like group. (**B**) Kaplan–Meier survival analysis comparing the overall survival of patients with the basal-like subtype in the high-risk (red) and low-risk (blue) groups for ATP1A1 genes. The high-risk group for *ATP1A1* was defined by its hypomethylation and high expression. The others were classified into the low-risk group. (**C**) The methylation level of the ATP1A1 CDS regions, cg11756509 and cg10891259, detected using the HM450K probe. (**D**) Overall survival of patients with the basal-like subtype, as determined by clustering of hypomethylation (blue) and hypermethylation (red) in two regions. (**E**) Kaplan–Meier analysis of relapse-free survival as a function of ATP1A1 expression in patients with basal-like breast cancer from the KM plotter database. (**F**) Relapse-free and distant metastasis-free survival in patients with basal-subtype breast cancer treated with chemotherapy stratified into groups with low (blue) and high (red) ATP1A1 expression based on the auto-selected optimal cut-off value. ** *p* < 0.01, *** *p* < 0.001 by log-rank tests.

**Figure 4 cancers-16-01666-f004:**
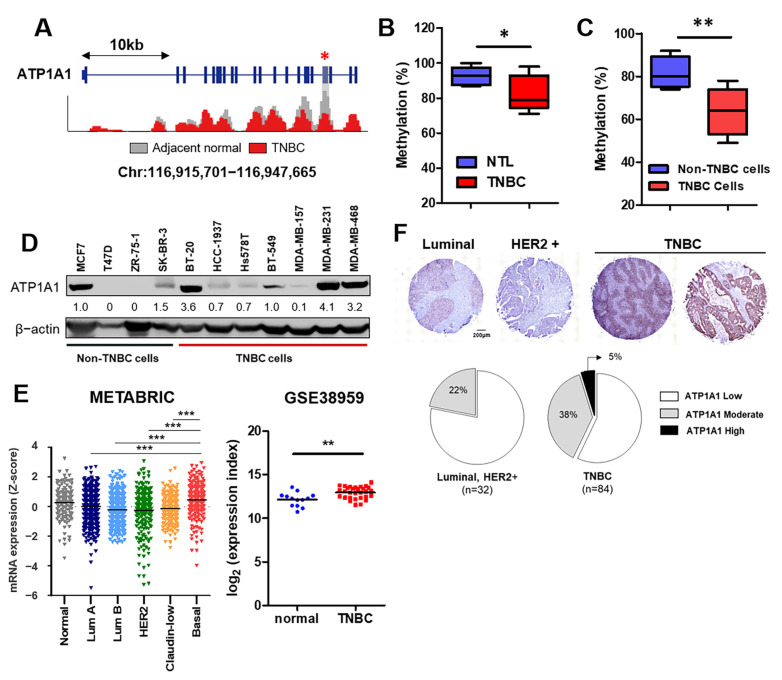
ATP1A1 is hypomethylated and overexpressed in TNBC tumors. (**A**) ATP1A1 methylation patterns in representative MBD-seq samples in TNBC and NTL. The red star indicates a significantly methylated region and its location for further analysis. (**B**) Methylation levels in NTL and TNBC tissues using MS-HRM in the selected region. (**C**) Methylation pattern in the selected region of non-TNBC (MCF7, T47D, SK-BR-3, and ZR-75-1) and TNBC (MDA-MB-157, MDA-MB-231, MDA-MB-468, BT-20, HCC-1143, HCC-1187, and HCC-1937) cell lines. (**D**) Expression of the ATP1A1 protein in breast cancer cell lines. (**E**) ATP1A1 mRNA expression in each PAM50 breast cancer subtype in the METABRIC datasets (left) and the expression of ATP1A1 in TNBC and mammary duct cells extracted from patients using GSE38959 (right). (**F**) Immunohistochemical results showing ATP1A1 expression in each TMA specimen from BR1202. * *p* < 0.05, ** *p* < 0.01, and *** *p* < 0.001. All experiments were performed independently three times.

**Figure 5 cancers-16-01666-f005:**
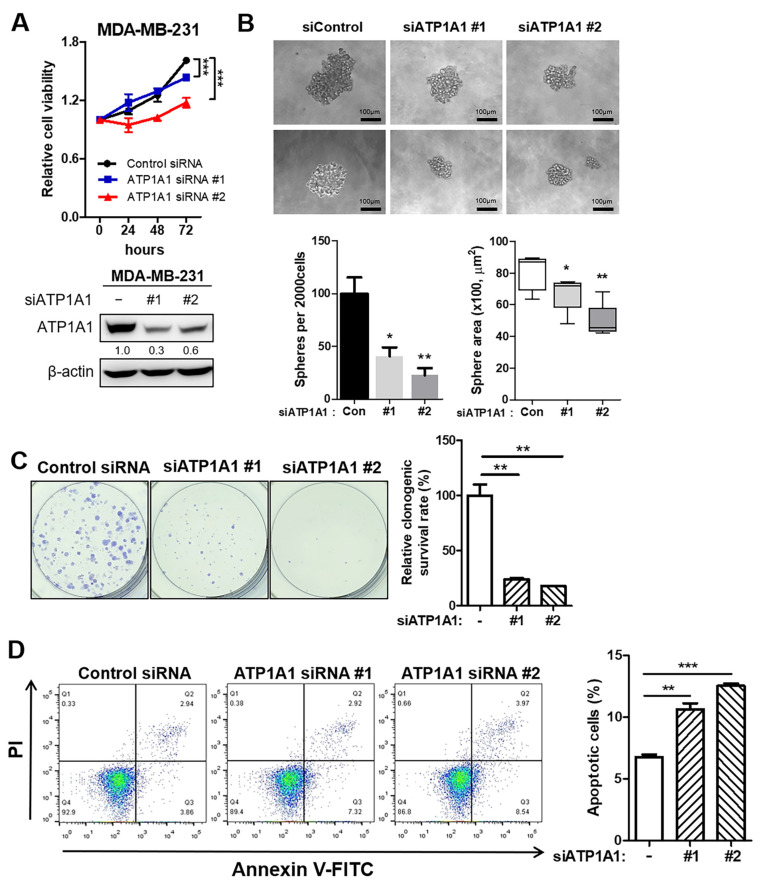
Inhibition of ATP1A1 expression mitigates aggressive tumor growth in TNBC. MDA-MB-231 cells were transfected with ATP1A1 siRNA #1 or #2 or control siRNAs. (**A**) Relative numbers of viable cells at the designated time points, as measured by WST-8 assays. (**B**) Numbers of mammospheres formed 10 days after transfection. Scale bar = 100 μm. Bar graph showing the number of spheres, and a box plot showing the sphere area of each group. (**C**) Representative images indicate colony formation in each group of cells. Bar graph showing the relative clonogenic survival rate. (**D**) Apoptotic cells were analyzed by flow cytometry. PI and FITC staining were used. Bar graph showing the apoptotic cell population in each group. * *p* < 0.05, ** *p* < 0.01, and *** *p* < 0.001. All experiments were repeated three times independently.

**Figure 6 cancers-16-01666-f006:**
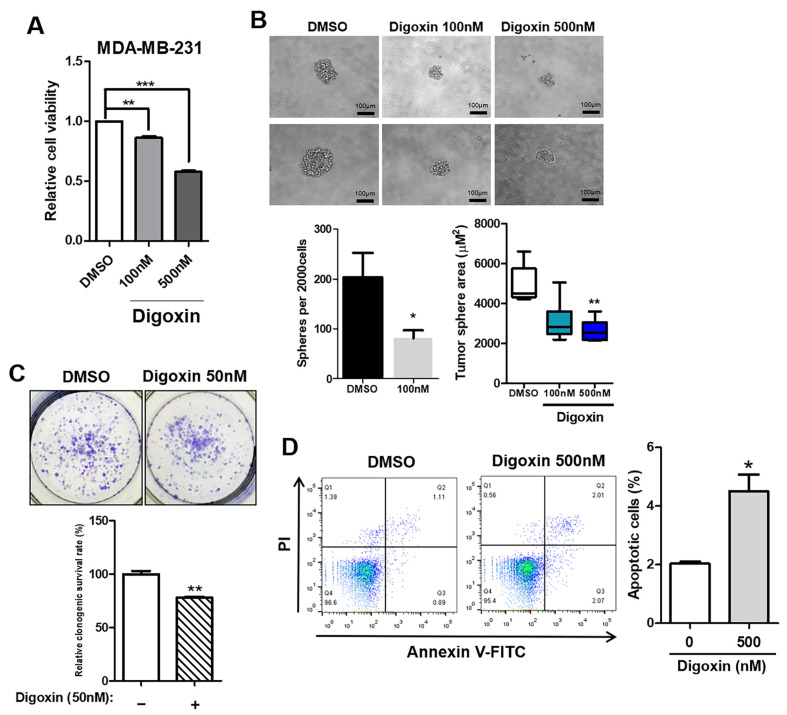
Inhibition of ATP1A1 activity mitigates TNBC progression in vitro. (**A**) Relative cell viability was measured by WST-8 assays after 24 h of treatment of MDA-MB-231 cells with DMSO or digoxin. (**B**) MDA-MB-231 cells were subjected to sphere formation assays for 9 days and then treated with digoxin (100 nM, 500 nM) or DMSO for 24 h. Representative images are shown. Scale bar = 100 μm. Numbers (upper) and areas (bottom) of tumor spheres in the DMSO- and digoxin-treated groups. (**C**) Representative images show colony formation in each group of cells. Bar graph showing the relative clonogenic survival rate under treatment with digoxin. (**D**) Apoptotic cells were analyzed by flow cytometry. PI and FITC staining were used. The bar graph shows the apoptotic cell population in each group. * *p* < 0.05, ** *p* < 0.01, and *** *p* < 0.001. All experiments were repeated three times independently.

## Data Availability

Data are contained within the article and Appendix A.

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
