# Peer review of "Hypomethylation of ATP1A1 Is Associated with Poor Prognosis and Cancer Progression in Triple-Negative Breast Cancer"

_cancers, 2024, doi:10.3390/cancers16091666_

Round 1

Reviewer 1 Report (Previous Reviewer 1)

Comments and Suggestions for Authors

The new manuscript offers more valuable insights into the exploration of ATP1A1 hypomethylation in TNBC. However, certain issues still need to be addressed before acceptance.

Figure 3: The shadows of each panel were different.  

Figure 5: The two siRNA target efficiencies showed inconsistency with the phenotype data (i.e., growth data); si#1 showed the highest knockdown efficiency, while si#2 groups were more significant than si#1 groups in phenotype data. How do you address these inconsistencies?

Sup Fig S5: Since forced ATP1A1 expression was successful in the system, it would be interesting to also add a sole ATP1A1 overexpression group to test the effect of ATP1A1 overexpression on TNBC cells. A WB blot to show the overexpression or qPCR data will be required.

Author Response

Reviewer 2 Report (Previous Reviewer 2)

Comments and Suggestions for Authors

The authors addressed all points and improved the manuscript accordingly. 

Author Response

Thank you for your positive comment. 

Round 2

Reviewer 1 Report (Previous Reviewer 1)

Comments and Suggestions for Authors

The choice of "Digoxin" as "targeting ATP1A1" also needs to be addressed.

Author Response

The choice of "Digoxin" as "targeting ATP1A1" also needs to be addressed.

: Thank you for this comment. As described in introduction and results, digoxin directly inhibits ATP1A1 function and has been clinically accepted in therapeutic agent for cardiovascular diseases. Interestingly, growing evidences suggest that digoxin also exerts antitumor effect in cancer. Therefore, by verifying whether the digoxin exhibits antitumor effect on TNBC, we tried to expand the application of the digoxin to diseases ranging from cardiac disease to breast cancer. We additionally described this content to the manuscript.  

This manuscript is a resubmission of an earlier submission. The following is a list of the peer review reports and author responses from that submission.

Round 1

Reviewer 1 Report

Comments and Suggestions for Authors

The manuscript provides an insightful exploration of ATP1A1 hypomethylation in TNBC, underscoring its potential role in prognosis and cancer progression.

Major Concerns:

  1. A workflow figure illustrating the process of identifying TNBC-specific DNA methylation would be beneficial for understanding the methodology more clearly.
  2. (Fig2E): The rationale behind comparing TNBC-specific methylation and mRNA expression patterns in a BRCA cohort with diverse subtypes, including Luminal A, B, HER2-enriched, basal-like, and normal-like samples, is unclear. A more focused comparison with TNBC samples would be more relevant.
  3. (Figures 3 and 4): The prognostic value of ATP1A1 is examined in a TCGA cohort, not specifically in a TNBC cohort. Since ATP1A1 expression is also high in MCF7 cells (a non-TNBC model), would also be interested to confirm the role of ATP1A1 in non-TNBC models.
  4. (Figure 5): Applying a methylation inhibitor in the experiments to provide more comprehensive insights.
  5. Regarding Digoxin's impact on ATP1A1 gene expression and its use at various concentrations (50nM, 100nM, 500nM) in different assays, a more detailed explanation is required. The rationale behind the selection of these specific concentrations and whether Digoxin's inhibitory effect on ATP1A1 can be counteracted by forced ATP1A1 over-expression should be addressed. This would aid in understanding the specific interactions and implications of Digoxin in the context of ATP1A1's role in TNBC.

Reviewer 2 Report

Comments and Suggestions for Authors

The study by Kim et al. describes specific DNA methylation and gene expression patterns in breast cancer tissue samples. They found ATP1A1 hypomethylation to be a possibly prognostic marker for TNBC and a potential therapeutic target. The results are well described, however, a few points should be adresse.

Abstract: ATP1A1 and CDS abbreviations should be introduced in similarity to MBD

  Introduction.  The definitinn of the intrinsic subtypes should be double-checked. Lumonal B type tumors are not necessarily HER2 poistive but show increased ki-67 index or higher expression of genes associated to proliferation and cell cycle.

Materials and Methods: 2000mMNaCl should be 2M.

Melting peaks were quantified by determining the area under the curve (AUC) using Image J software. This is confusing since AUC might refer to sensitivity and specificity of a test in terms of ROC analysis. Did the authors mean DNA quantification by using the integration of the melting peak curve?

Immunohistochemical analysis: What specimens were used for positive and negative control stainings?

How was the apoptosis rate calculated?

Results:

Fig 1. Caption: gray should be grey. A) The heatmap is is difficult to read.

Handling of patient derived data like overall survival from the databases TCGA or GEO should be described in the Material and Methods section. A table describing the “cohort” would be interesting (age, therapies, TNM).

What does NTL stand for?

Discussion: Considerimg the comprehensive methodologies interesting results, the discussion is rather short, but straight to the point. In order to conclude wether ATP1A1 is truly prognostic, more clinical data such as therapies would be interesting. The authors should discuss possible clinicla trials.

Reviewer 3 Report

Comments and Suggestions for Authors

Comments on the manuscript, it is an interesting article where the authors study hypomethylation of ATP1A1 associated with poor prognosis and cancer progression in triple-negative breast cancer. some minor comments are described below

Show figure 1A in greater definition

Expand the discussion and discuss each of the results found, propose a mechanism that explains that hypomethylation and high expression of the ATP1A1 gene, especially exon 19, are associated with the poor survival prognosis in TNBC.
